# Investigation of the Film-Cooling Performance of 2.5D Braided Ceramic Matrix Composite Plates with Preformed Hole

**Chenwei Zhao, Zecan Tu and Junkui Mao ***

Jiangsu Province Key Laboratory of Aerospace Power System, College of Energy and Power Engineering, Nanjing University of Aeronautics and Astronautics, Nanjing 210016, China; zcwphd@nuaa.edu.cn (C.Z.); tzc@nuaa.edu.cn (Z.T.)
* Correspondence: mjkpe@nuaa.edu.cn; Tel.: +86-025-8489-6995

**Abstract:** The film-cooling performance of a 2.5D braided ceramic matrix composite (CMC) plate with preformed holes was numerically studied. Four numerical models containing braided structures were established: one model with film-cooling holes preformed through fiber extrusion deformation (EP-Hole), one model with film-cooling holes directly woven through fibers (WP-Hole), and two models with directly drilled holes (DP-Hole1,2). Besides, the influence of the ratio between the equivalent thermal conductivities on the axial and radial directions of fiber $Kr$ was investigated. The results show that the preformed holes have better performance in controlling the thermal gradient with the increase of $Kr$. The maximum thermal gradient around the DP-Hole is significantly higher than that of the WP-Hole and EP-Hole, and the maximum relative variation reaches 123.3%. With $Kr$ increasing from 3.32 to 13.05, the overall cooling effectiveness on the hot-side wall decreases for all models, by about 10%. Compared with the traditional drill method, the new preformed film-cooling hole studied in this paper can reduce the temperature and the thermal gradient in the region around the holes.

**Keywords:** film cooling; anisotropic thermal conductivity; 2.5D braided composite; preformed hole

## 1. Introduction

With the continuous increase of the thermal load on the hot components of advanced propulsion systems [1,2], the temperature resistance requirements for the materials of the hot components have also been continuously improved. Braided ceramic matrix composite (CMC) has excellent heat resistance and mechanical properties, which makes it a popular candidate material for high-temperature components of advanced propulsion systems [3–7]. After years of research and development, braided CMC hot components have been applied by more and more aero-engine manufacturers [8–10].

However, the braided fibers and matrix of CMC could be oxidized in high-temperature environments, and the literature [11] shows that the oxidation rate of CMC increases with the increase of environmental temperature, especially in ultra-high temperature environments (>1500 °C). The oxidation loss rate of Cf/SiC composite will increase sharply. The oxidation of CMC will greatly reduce its mechanical properties [12–14]. The GE Company's research [15] showed that the ultimate strength degradation of CMC reinforced with Hi-Nicalon fibers was less than 10% after being exposed to the air at 1200 °C for 4000 h, while the ultimate strength degraded 30% after exposure to air at 1315 °C for 1000 h. Unal's [16] research on SiC/SiC composites showed that after SiC/SiC composite samples were exposed to a dry oxygen environment at 1400 °C for 50 h, the fracture stress of the oxidized specimens decreased by about 50%. Therefore, the CMC components still need cooling structures to protect them from oxidation.

Film cooling is one of the most widely used cooling technology for hot components in propulsion systems due to its simple structure and high cooling effectiveness. It has already been applied in CMC hot components [17], such as the CMC turbine vane and

CMC combust chamber. Prediction of the cooling efficiency and the internal temperature is necessary for the design of CMC components with high temperatures. To study the film-cooling performance of CMC hot components, some scholars currently research CMC plates. Z.C. Tu [18,19] studied cooling performance with circular film-cooling holes of a unidirectional fiber-toughened CMC plate. The study showed that the anisotropy thermal conductivity would affect the heat transfer process inside the film cooling plate. In the direction of toughening, the heat transfer capacity is strengthened, which in turn will affect the overall cooling efficiency and temperature distribution uniformity of the film-covered wall. However, the influence of the microstructure of CMC on film-cooling performance was not considered, especially the effect of fiber on heat transfer. Actually, braided fibers will greatly affect the internal heat transfer mechanism of the material [20]. X. Zhao [21] studied the influence of the relative position of the hole and the braided fiber on the film-cooling performance of a 2.5D braided CMC plate. The results show that the interference between the film-cooling hole and the fiber will affect the film cooling and the temperature field distribution of the plate. Therefore, the change of braided structure around film-cooling holes results in significant changes in the heat transfer path, temperature field, and thermal gradient distribution around the film-cooling hole.

The use of film-cooling technology requires discrete film-cooling holes drilled in the component [22,23], which may cause stress concentration around the hole. For CMC, its matrix is ceramic with poor toughness, and the braided fibers have a significant impact on CMC's overall mechanical properties. Drilling will cut off the internal braided fibers, resulting in serious stress concentration and greatly weakening the structural strength [24,25]. Besides drilled holes, the required holes can also be created by fiber braiding. Compared with the drilled structure, the mechanical properties of the structure with preformed holes are greatly improved, because the braided fibers around the holes are not damaged. Z. G. Liu [26] carried out a uniaxial static tensile test on a lug with braided and drilled holes, and the research results showed that the bearing capacity of the three-dimensional and five-directions braided lug with the preformed hole was 520% higher than that of the drilled lug.

In general, there are two ways to manufacture preformed holes: one makes braided-preformed holes (BP-Hole) through the weaving process [26], and the other one makes extruded preformed holes (EP-Hole) by extruding the braided fibers [27]. The research in [15] has shown that the interference between the film-cooling hole and the braided fiber will directly affect the film-cooling performances of the 2.5D braided plate. Both EP-hole and WP-hole will change the braided structure around the film-cooling hole. Therefore, the influence of the preformed hole on the film-cooling performance needs to be investigated.

Additionally, the anisotropic thermal conductivity of the braided fiber also directly affects the heat transfer and temperature distribution inside the CMC plate [20], resulting in changes to the film-cooling performance under the same braided structure. Different braided fibers have different thermal conductivity performances, such that the equivalent thermal conductivity of the fibers are different and the axial-to-radial thermal conductivity ratio of the fibers is also inconsistent [28–31]. The axial thermal conductivity of Hi-Nicalon fiber (Type S) is 18 W/(m·K) [28], but the axial thermal conductivity of T800 fiber is only 5 W/(m·K) [28]. The axial-to-radial thermal conductivity ratio of C-T700 is 15.7 [30], but the axial-to-radial thermal conductivity ratio of SiC fiber is only 6.52 [21]. Therefore, the influence of the axial-to-radial thermal conductivity ratio of fibers on the film cooling performance needs to be investigated too.

Accordingly, in this paper, the influence of the preformed hole on film-cooling performance over a 2.5D braided composite plate is studied. Four CMC plate models, including EP-hole, WP-hole, and two drilling holes (DR-Hole), were established and simulated. Subsequently, the film-cooling performance of the above four models with different axial-to-radial thermal conductivity ratios of fibers is also studied.

## 2. Numerical Methodology

### 2.1. Numerical Models

Figure 1a shows the numerical model of 2.5D braided CMC plate with a film-cooling hole, which consists of four parts: film-cooling hole, mainstream flow region, coolant flow region, and 2.5D braided CMC plates. The diameter (D) and the injection angle of the film cooling hole are 2.4 mm and 90°, respectively. The mainstream flow region and coolant flow region extend 7.5D upstream from the center of the film-cooling hole and 15.5D downstream. The span-wise spacing of the film-cooling hole is 3D. The size of these two regions is 23D × 3D × 8D (length × width × height). Figure 1a,b show their detailed dimensions.

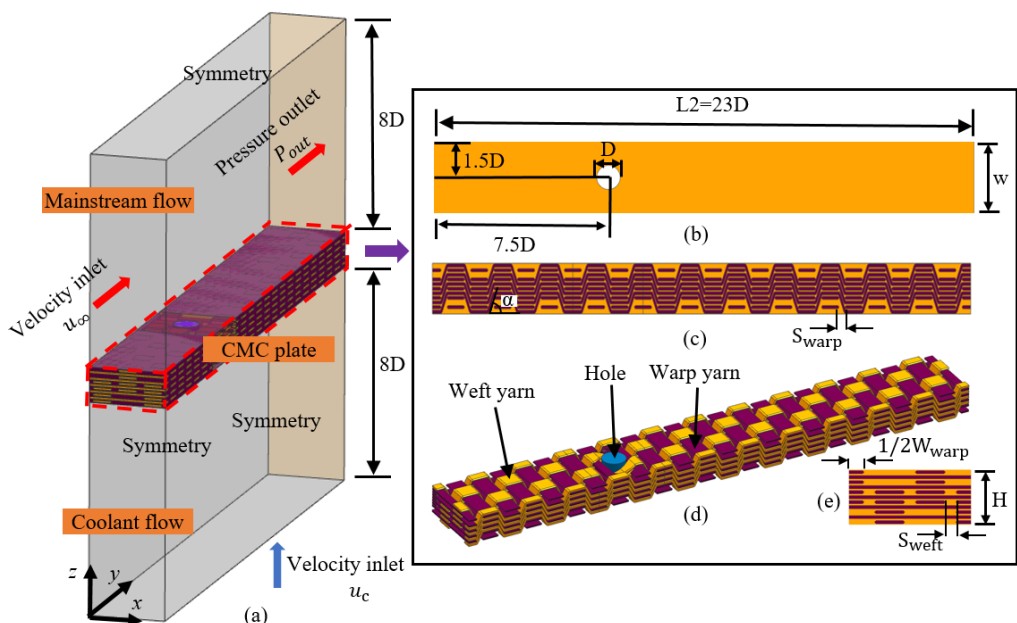

**Figure 1.** Numerical models of the film cooling over a 2.5D braided CMC plate: (**a**) schematic diagram of calculation model; (**b**) hot-side wall; (**c**) lateral view of the plate; (**d**) fiber structure of the plate; (**e**) the left view of plate.

The 2.5D braided CMC plate is composed of matrix and braided fibers. The geometric dimensions of the braided structure in the *y-x* plane and *y-z* plane are given in Figure 1b–e. The thickness (H) of the plate is 3.3 mm in the *z*-axis direction. The length of the plate (L) in the *x*-axis direction is 23D. In the *y*-axis direction, the width of the plate (W) is 3D.

For braided structures, the width of weft yarns ($W_{weft}$) is 1.8 mm, and the distance between two adjacent weft yarns ($S_{weft}$) is 0.6 mm. The width of warp yarn ($W_{warp}$) is 1.8 mm, and the gap between two adjacent warp yarns ($S_{warp}$) is 0.6 mm. The braiding angle ($\alpha$) is 70.5°.

The numerical model in this paper involves four physical models for different manufacturing processes of holes. The difference lies in the changes of fiber-braiding structure around the film-cooling holes. Generally, the manufacturing processes of holes have two different methods, including the drilled hole and the preformed hole. For the drilled hole, firstly, the preform is woven by fibers, and the preform is compounded with the matrix material to obtain CMC material. Then the holes are drilled directly into the finished braided material, and the film-cooling hole can be obtained by breaking the fibers and removing the matrix at the corresponding position. The manufacturing schematic diagram of the drilled hole is shown in Figure 2a. For the preformed hole, firstly, the filler of the required film-cooling hole is manufactured, and the filler is placed at the position of the film-cooling holes. The preform fiber directly bypasses the film-cooling hole filler to preform the film-cooling holes. Then, the preform and the matrix material are composite

sintered to make CMC components; in the manufacturing process, the film-cooling holes' filler can be directly decomposed in the sintering process, such as with graphite and other materials. A special solution also can be used to decompose the film-cooling hole filler after the CMC component is sintered without damaging the CMC component. The shape of the preformed hole is guaranteed by the filler. In this paper, the filler of the film-cooling hole is a straight cylinder. The manufacturing schematic diagram of the preformed hole is shown in Figure 2b.

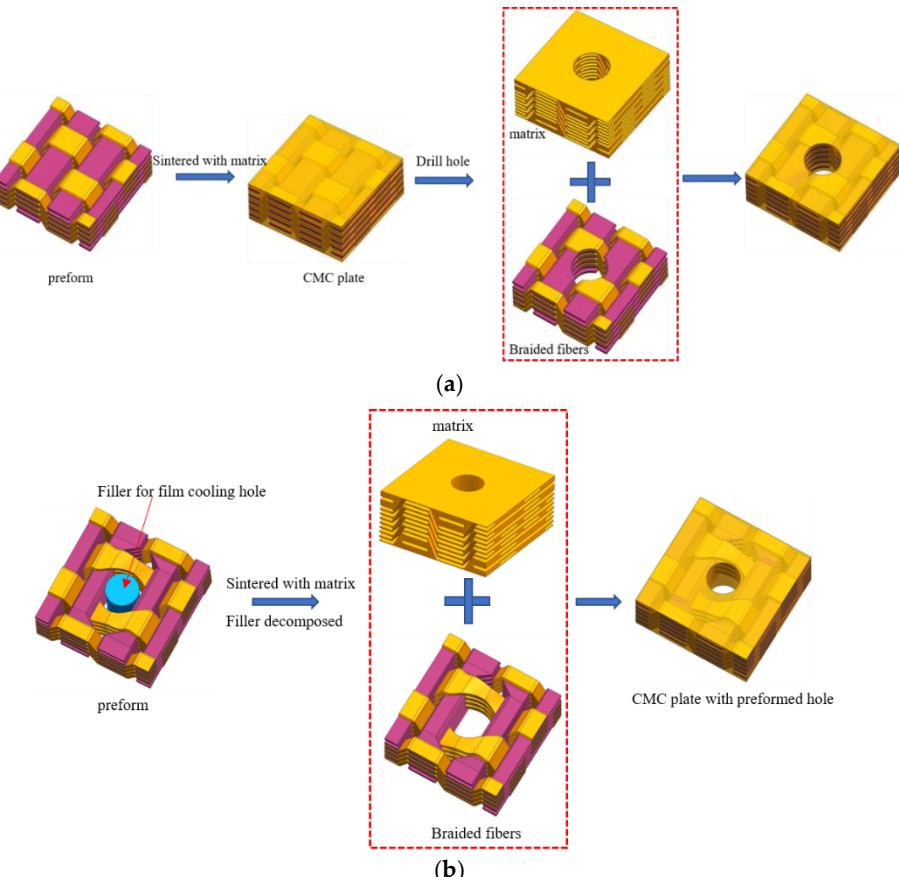

**Figure 2.** The different manufacturing schematic diagrams for film-cooling holes: (**a**) the manufacturing schematic diagram of the drilled hole; (**b**) the manufacturing schematic diagram of the preformed hole.

In general, two preformed holes with different braided structures are investigated, and the corresponding drilled holes are also studied to provide the reference objects for preformed holes. Finally, the numerical models with different holes built in this paper include: (1) the numerical model with preformed film-cooling holes through fiber extrusion deformation (EP-Hole) and the corresponding drilled model (DR-Hole1), the specific model details of which are shown in Section 2.1.1; and (2) the numerical model with preformed film-cooling holes directly woven through fibers (WP-Hole) and the corresponding drilled model (DR-Hole2), the specific model details of which are shown in Section 2.1.2.

### 2.1.1. Numerical Models with EP-Hole and DR-Hole 1

When the preform is braided, there will be a natural gap with the size of $S_{wrap} \times S_{weft}$ between the fibers, as shown in Figure 3a. The EP-Hole is based on the natural gap between the braided yarns, which is enlarged to form a film-cooling hole. The EP-Hole is located between two weft yarns and two warp yarns. When weft and warp yarns are woven to the filling position of the hole, the fiber is squeezed by the filler, deforming and bypassing the filler at the same time. When the preform is sintered with the matrix, the ceramic matrix

will fill the gaps between the preform fibers. At the same time, due to the existence of filler, the matrix material will not fill the filler area, and eventually leave a film hole with the same shape as the filler.

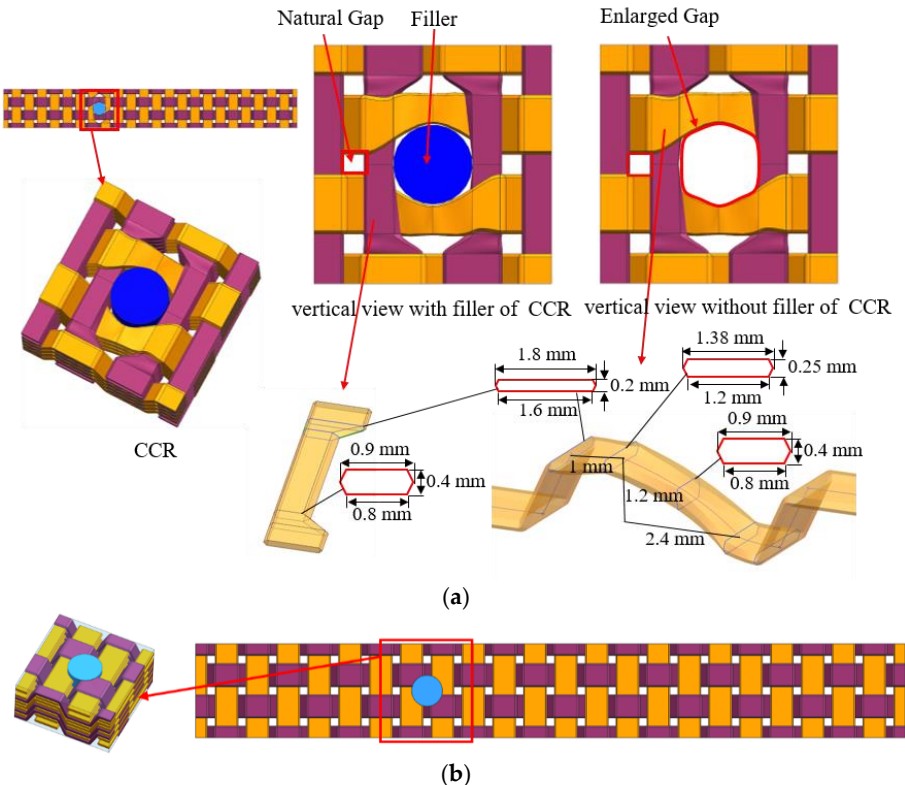

**(a)**

**(b)**

**Figure 3.** Internal structures of Geo 1 and 2: (**a**) Geo1, (**b**) Geo2.

In the numerical simulation, it is necessary to establish a model containing the extruded fiber. The characterization method of the fiber's squeeze is shown as the following: it is ensured that the cross-sectional area remains unchanged when the fiber is deformed. The cross-section is swept along the guide line to obtain the fiber entity, and finally, the volume of the fiber is unchanged. The detailed modeling process is shown in Figure 4. Based on the above modeling method, the CMC plate geometric model with EP-hole (Geo1) is established; the detail size is shown in Figure 3a. The weft yarns and warp yarns near the hole are squeezed. The weft section changes from 1.8 mm × 0.2 mm to 1.38 mm × 0.25 mm, then to 0.9 mm × 0.4 mm, and the warp section changes from 1.8 mm × 0.2 mm to 0.9 mm × 0.4 mm. To compare the influence of the structure with EP-hole and DR-Hole on film-cooling performance, a comparable geometric model with DR-Hole 1 (Geo2) is established, as shown in Figure 3b.

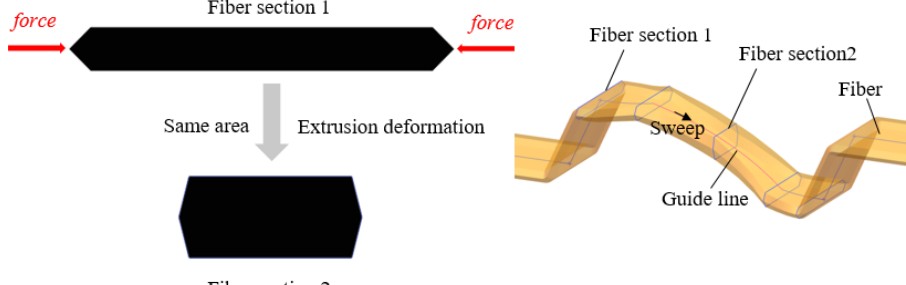

**Figure 4.** Schematic diagram of fiber deformation and modeling.

### 2.1.2. Numerical Models with BP-Hole and DR-Hole 2

In addition to the preformed film-cooling holes created by extruding and braiding fibers, like geo1, a new method for BP-Hole is used in our research. In this method, when the weft is woven to the film-cooling hole, it is retraced in the opposite direction at the same time. When the weft is retraced, the warp is pulled to move to both sides of the film-cooling hole to directly braid preformed holes (BP-Hole). The warp without weft traction is deformed by the extrusion of the filer. The detailed size of the geometric model with BP-Hole is shown in Figure 5a. The weft section is 1.8 mm × 0.2 mm, and the warp section changes from 1.8 mm × 0.2 mm to 2.12 mm × 1 mm. To compare the effect of the structure with BP-Hole and DP-Hole on the film-cooling performance, another model with the drilled hole (DR-Hole 2 (geo4)) is established as the comparison model, as shown in Figure 5b.

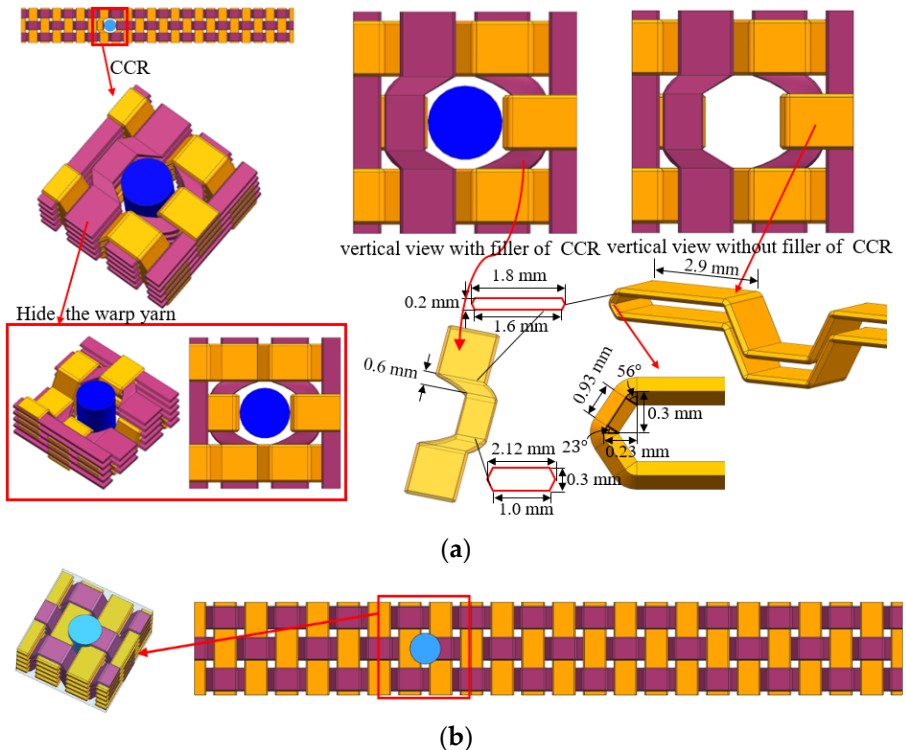

**Figure 5.** Internal structures of Geo 3 and 4: (**a**) Geo3, (**b**) Geo4.

### 2.2. Calculation Coordinates for Anisotropic Thermal Conductivity

As described previously, the thermal conductivity ($k_{ij}$) of braided fibers is anisotropic, and the thermal conductivity along the fiber bundle direction is higher than that in the direction perpendicular to the fiber bundle. The anisotropic thermal conductivity of braided fibers can be expressed by a second-order tensor.

$$k_{ij} = \begin{bmatrix} k_{xx} & k_{xy} & k_{xz} \\ k_{yx} & k_{yy} & k_{yz} \\ k_{zx} & k_{zy} & k_{zz} \end{bmatrix} \tag{1}$$

At present, based on the assumption of homogenization, researchers can use numerical simulation or experimental measurement to obtain the thermal conductivity of braided

fiber in three main directions (along the fiber bundle direction and perpendicular to the fiber bundle direction), and then Equation (1) can be simplified as a second-order diagonal tensor.

$$k_{ii} = \begin{bmatrix} k_{xx} & 0 & 0 \\ 0 & k_{yy} & 0 \\ 0 & 0 & k_{zz} \end{bmatrix} \tag{2}$$

As shown in Figure 2a, during 2.5D braiding, the weft yarns will be deflected in space, which will make the main direction of the weft yarns' heat conduction mismatch with the global calculation coordinate system. The preformed holes will cause the weft yarn around the film cooling hole to deflect. The deformation also causes the main direction of the heat conduction of the braided yarn to not match the global coordinate system. When the global calculation coordinate system selection is inconsistent with the main direction of heat conduction, the internal temperature field and heat transfer path of the CMC plate cannot be accurately calculated. Therefore, the curvilinear coordinate system is applied in our study to make the main direction of heat conduction constant with the path of the braiding yarn. Figure 6 shows the global coordinates and curvilinear coordinates used in this paper, in which the global coordinates are used for the fluid region and matrix region, and the curvilinear coordinates are used for the braided fibers, where the *x* direction of the curvilinear coordinates is along the tangent direction of the fiber direction. The anisotropic thermal conductivity of CMC can be calculated accurately by combining Equation (2) with curvilinear coordinates.

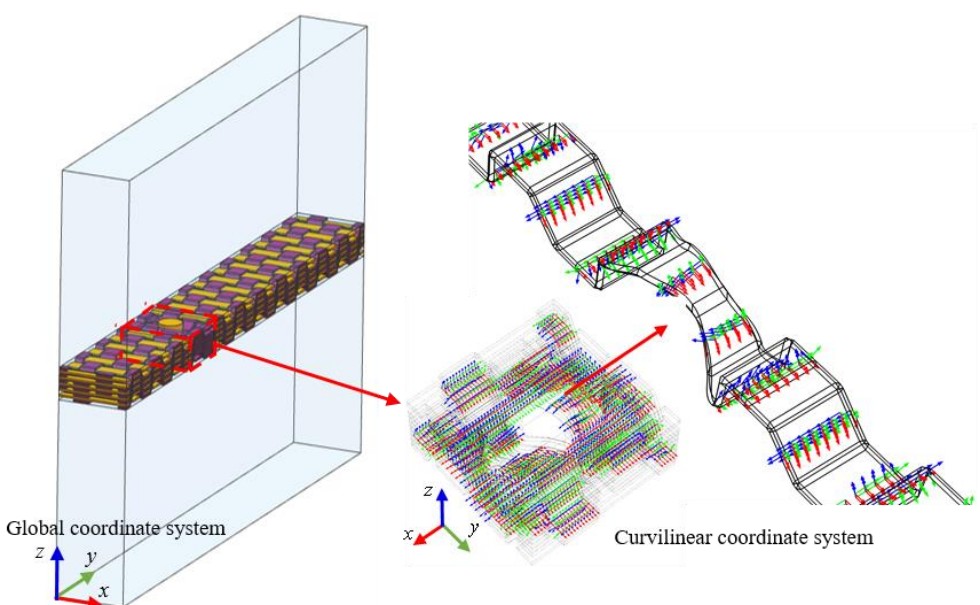

**Figure 6.** Schematic diagram of calculated coordinates.

### 2.3. Boundary Conditions and Parameter Definitions

As shown in Figure 1a, the mainstream inlet is the velocity inlet, and the mainstream outlet is the pressure outlet. At the inlet, $u_\infty$ is 20 m/s, and $T_\infty$ is 360 K, and the turbulence intensity is 0.2%. The static pressure ($P_{out}$) at the outlet is 1 atm. For the coolant flow, the inlet is also a velocity inlet, where $u_c$ is 0.04823 m/s, and $T_c$ is 300 K. The left and right boundaries of the calculation domain in the *x* direction are symmetrical boundaries, and the remaining boundaries are non-slip adiabatic walls. Based on the above boundary conditions, the blowing ratio (Br) of the numerical simulation of film cooling in this paper is 0.25.

The blowing ratio (Br) is defined as:

$$\text{Br} = \frac{\rho_c u_c}{\rho_\infty u_\infty} \tag{3}$$

where $\rho_\infty$ and $\rho_c$ are the densities of the mainstream and the coolant at the inlet of the film-cooling hole.

According to the data of 2.5D braided composite provided by the material supplier, the thermal conductivity of the matrix is 0.2 W/(m·K), the axial equivalent thermal conductivity ($k_{xx}$) of the braided fiber is 9.66 W/(m·K), and the radial equivalent thermal conductivities ($k_{yy}$ and $k_{zz}$), which are perpendicular to the direction of the fibers, are both 1.48 W/(m·K). As mentioned above, the film-cooling performance of 2.5D braided composite plates was studied when the ratio of axial-to-radial thermal conductivity (Kr) was 3.26, 6.52, and 13.05, respectively. The detailed thermal conductivity values are shown in Table 1.

**Table 1.** Thermal conductivities of the material.

| Serial | Thermal Conductivity in Axial of the Fiber Bundles ($k_{xx}$)/W/(m·K) | Thermal Conductivity in Radial of the Fiber Bundles ($k_{yy}$ and $k_{zz}$)/W/(m·K) | Thermal Conductivity of Matrix ($k_m$)/W/(m·K) | Kr |
|---|---|---|---|---|
| 1 | 4.83 | 1.48 | 0.2 | 3.26 |
| 2 | 9.66 | 1.48 | 0.2 | 6.52 |
| 3 | 19.32 | 1.48 | 0.2 | 13.05 |

The ratio of the axial and radial thermal conductivity (Kr) is defined as:

$$\text{Kr} = \frac{k_{xx}}{k_{yy}} \tag{4}$$

To analyze the film-cooling performance, the overall cooling efficiency $\eta$ and average dimensionless temperature $\delta$ are introduced, as shown in Equations (5) and (6):

$$\eta = \frac{T_\infty - T_w}{T_\infty - T_c} \tag{5}$$

$$\delta = \frac{T_{\text{aver}} - T_c}{T_\infty - T_c} \tag{6}$$

where $T_\infty$ and $T_c$ are the temperatures at the mainstream inlet and the coolant inlet, respectively; $T_w$ is the temperature on the hot-side wall; and $T_{\text{aver}}$ is the average temperature in the selected area.

To analyze the temperature gradient, relative temperature gradient $\varepsilon$ is introduced, as shown in Equation (7):

$$\varepsilon = \frac{T_{\text{grad}} H}{T_\infty - T_c} \tag{7}$$

where $T_{\text{grad}}$ is the temperature gradient.

In this study, the film-cooling effectiveness of the hot-side wall is firstly studied. To analyze the cooling performance of the hot-side wall clearly, two special lines (as shown in Figure 7a) are introduced, namely, line 1 (centerline of the outflow) and line 2 (X/D = 4). Secondly, to study the influence of the different braided structures of the film-cooling holes on the temperature field and heat transfer path of the plate, a cuboid characteristic region (CCR) centered on the center of the film-cooling hole is selected; the size of the region is 3D × 3D × H (length × width × height). Finally, to study the temperature and heat transfer mechanism of film-cooling 2.5D braided composite plate, four characteristic sections were selected as the analysis objects. There are three characteristic sections along the *x*-axis direction, which are section 1 (X = 0.5D), section 2 (X = 0), and section 3 (X = −0.5D). There is a characteristic section 4 (Y = 0) along the *y*-axis direction. The selected characteristic region and the location of the characteristic sections are shown in Figure 7b.

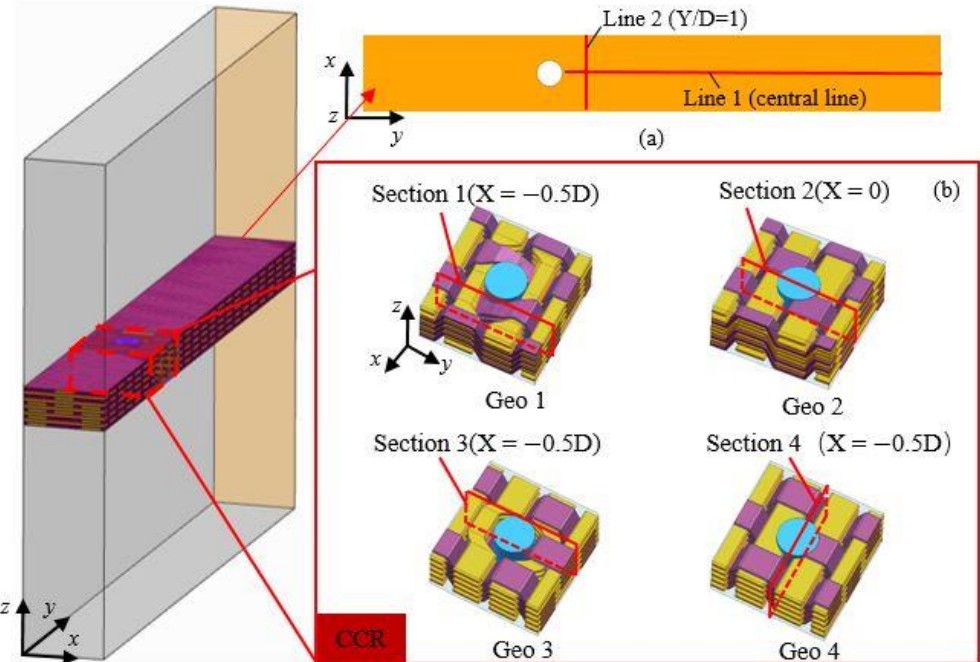

**Figure 7.** Diagram of the locations of characteristic sections and feature lines: (**a**) Schematic diagram of the line location; (**b**) Schematic diagram of the section location.

### 2.4. Grid System

Mixed grids are used for numerical simulation calculations. In the solid region, a tetrahedral grid is used, and then in the mainstream region, the coolant flow region and the film-cooling hole region, the grids on the top and bottom surfaces of the plate are swept to obtain the swept grids. To ensure that the calculation results are independent of the number of grids, this paper divides four sets of grids with the total number of 1,678,344, 3,473,632, 5,386,566, and 8,423,522, respectively, for the verification of grid independence. Figure 8a shows the overall cooling effectiveness under different grid numbers on line 1. The results show that when the number of grids is changed from 5,386,566 to 8,423,522, the change in the overall cooling efficiency is less than 1%. Therefore, a grid with 5,386,566 nodes was used in the simulation, as shown in Figure 8b. In the mainstream flow and coolant flow region, the maximum size of the grid is 0.33 mm, the minimum size is 0.15 mm, and the element size growth rate is 1.1. The height of the first layer of the grid is $1.5 \times 10^{-2}$ mm, with a total of 15 layers, and the stretch ratio is 1.1. In the film-cooling hole, the maximum size of the grid is 0.1 mm and the minimum size is 0.04 mm, with an element size growth rate of 1.05. For 2.5D braided CMC plates, the maximum size of the grid is 3 mm and the minimum size is 0.1 mm, with an element size growth rate of 1.4.

### 2.5. Grid System

The simulation was carried out with COMSOL Multiphysics 5.4 [32], COMSOL Inc, Stockholm, Sweden. In the CFD (Computational fluid dynamics) model verification, we used the standard k-ε model and the modified k-ω model provided by the software [33]. Based on the models in [34] and [35], adiabatic film cooling simulation is carried out. Figure 9 shows the cooling effectiveness on the centerline of the film-cooling hole downstream and the span-wise line (Y/D = 1) when these two models are used to calculate, respectively. At the same time, the experimental results of the literature [35] and the calculation results of the SSTA model are used in the literature [34].

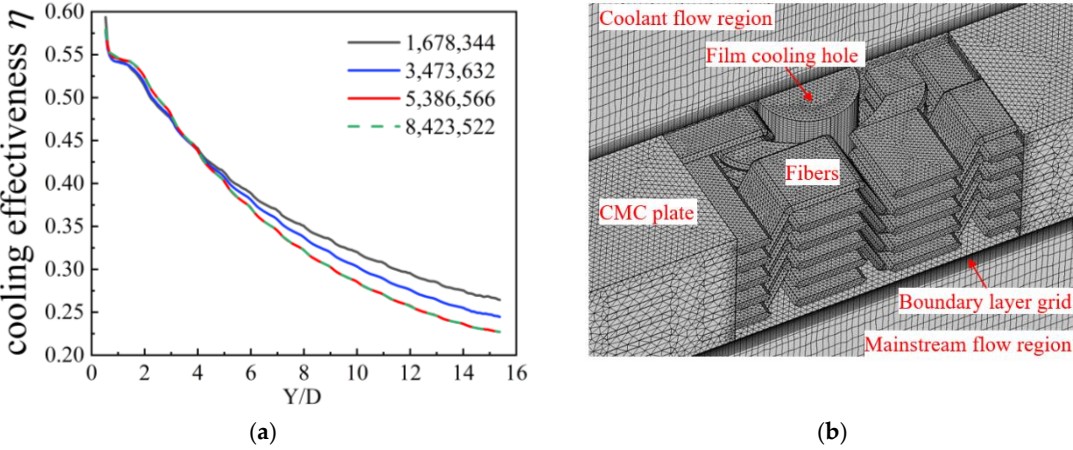

(**a**)                                                                      (**b**)

**Figure 8.** Generation of the grid: (**a**) cooling effectiveness on line 1 with the different grid; (**b**) schematic diagram of the grid.

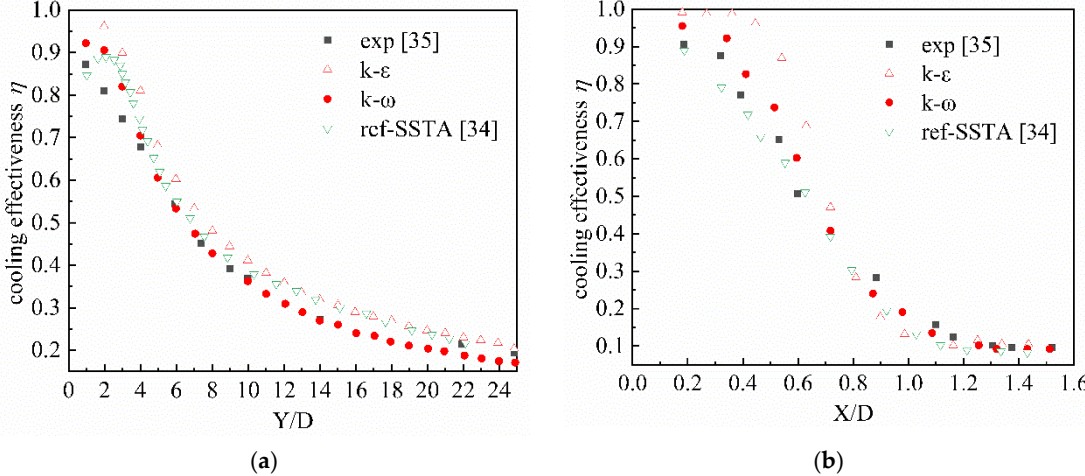

(**a**)                                                                      (**b**)

**Figure 9.** Validation of the computational method: (**a**) central line, (**b**) span-wise line.

On the centerline, the calculation results of the k-ε model and k-ω model are compared with the calculation results of the SSTA model, and it can be found that the cooling effectiveness of the downstream centerline of the film-cooling hole obtained by the k-ε model and k-ω model decreases monotonously. Consistent with the experimental rules, and in the calculation results of the ref-SSTA model, the cooling effectiveness peaks at about 2D–3D downstream of the film-cooling hole. Therefore, the calculation results of the k-ε model and the k-ω model are more in line with the experimental rules, and compared with the k-ε model, the cooling effectiveness calculated by the k-ω model is closer to the experimental value; the average relative error compared with the experiment data is 3.98%.

On the span line (Y/D = 1), the k-ε model, k-ω model, and SSTA model have a certain gap between the span-wise cooling effectiveness distribution and the experimental value, but in general, the calculation results of the k-ω model and SSTA model are in good agreement with the experimental values. The average relative error on the span-wise line is 5.24% for the k-ω model.

Additionally, to verify the simulating accuracy of the overall cooling effectiveness over the 2.5D braided plate, the simulation is carried out and compared with an overall film-cooling experiment on 2.5D braided plate by our research group [21]. The manufacturing processing of the film-cooling hole of the composite material test piece in reference [21] is consistent with that of the DR-Hole in this paper. The modeling method introduced in Section 2.1 is adopted in this section to model the film-cooling hole and conduct a numerical simulation. Figure 10 shows the calculation results and the experimental data

of [21] on line 1 under the corresponding working conditions. It can be seen from Figure 10 that the two results are similar in the area near the outflow direction of the film cooling hole (Z/D < 4), and the calculated value is lower in the far downstream area of the film-cooling hole (Z/D > 10). The average relative error between the calculated result and the experimental result is 4.39%.

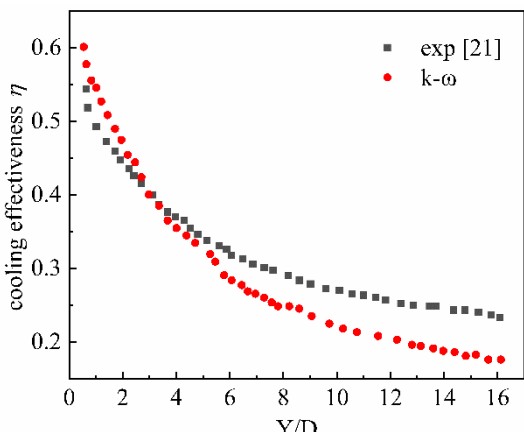

**Figure 10.** Validation of the computational method for overall cooling effectiveness.

Considering the above results and analysis, and according to previous research [36], the k-ω model has good accuracy in predicting the overall film-cooling effect. The k-ω model is used for subsequent numerical calculations provided by the CFD module in COMSOL Multiphysics 5.4 [32], COMSOL Inc, Stockholm, Sweden.

*2.6. Operating Conditions*

Based on the above four models, to explore the film-cooling characteristics of the four cooling structures under different Kr, 12 cases are simulated, as shown in Table 2. The film-cooling performances of EP-Hole at different Kr are calculated in cases 1, 5, and 9; WP-Hole in cases 3, 7, and 11; DR-Hole 1 in cases 2, 6, 10; and DR-Hole 2 in cases 4, 8, 12, respectively. By comparing cases 1–4, the effects of different hole-forming methods on the film-cooling performances of 2.5D braided CMC plate can be obtained. By comparing the results of cases 1–4, cases 5–8, and cases 9–12, the overall film-cooling performances of 2.5D braided CMC plate with different hole-forming methods and different Kr coupling effects can be obtained.

**Table 2.** The operating conditions of different cases.

| Case | 1 | 2 | 3 | 4 | 5 | 6 | 7 | 8 | 9 | 10 | 11 | 12 |
|------|---|---|---|---|---|---|---|---|---|----|----|----|
| Kr | | 6.52 | | | | 3.26 | | | | 13.05 | | |
| Geo | Geo1 | Geo2 | Geo3 | Geo4 | Geo1 | Geo2 | Geo3 | Geo4 | Geo1 | Geo2 | Geo3 | Geo4 |

## 3. Results and Discussions
### 3.1. Influence of Different Film-Cooling Holes on Film-Cooling Performance
### 3.1.1. Influence on the Overall Cooling Effectiveness of the Hot-Side Wall

Figure 11a shows the $\eta$ distribution of the hot-side wall under different film-cooling holes (cases 1–4). The overall cooling effectiveness at different positions on the hot-side wall is not a smooth transition but shows obvious fluctuation related to braided structure. The fluctuation of $\eta$ is more obvious in the upstream (−5D–0D) of the film hole, and the maximum $\eta$ fluctuation is 14.3%. In the film coverage region (0 < Y < 3D) and non-film coverage region downstream of the film-cooling hole, the fluctuation of $\eta$ is still strong, but weaker than that of the upstream region, and the maximum fluctuation of $\eta$ is 12.8%. In the

film coverage region (3D < Y < 15.5D) downstream of the film-cooling hole, the fluctuation of $\eta$ is the weakest, and the maximum value is only 7.4%.

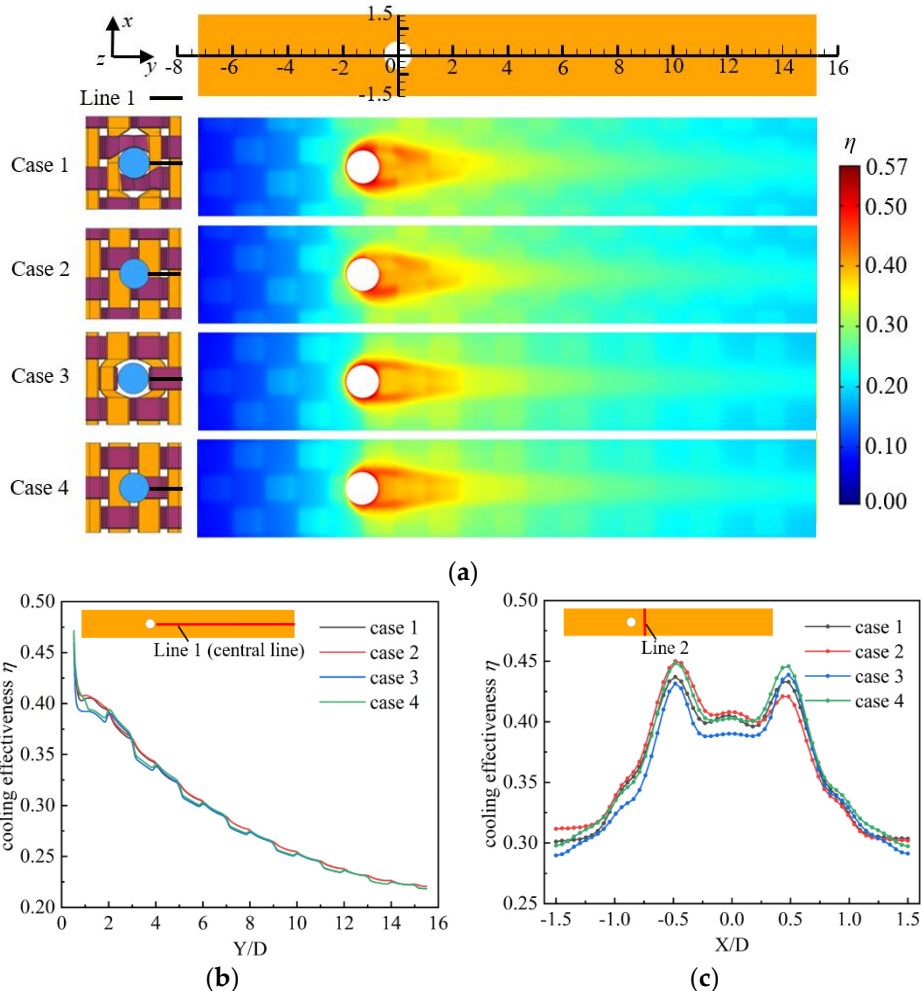

**Figure 11.** Temperature fields on hot-side wall and cooling efficiencies of cases 1–4: (**a**) distribution of $\eta$ on hot-side wall; (**b**) cooling effectiveness on line 1; (**c**) cooling effectiveness on line 2.

The difference of $\eta$ on the hot-side wall is caused by different internal structures and conjugate heat transfer with the cooling film. The braided fiber inside the plate affects the heat transfer path of the conjugate heat transfer process with film cooling. On the one hand, the thermal conductivity of fiber and matrix is different (kr = 6.52 for cases 1–4), which will make the fiber and matrix show different heat transfer characteristics. On the other hand, the axial thermal conductivity of the braided fiber is significantly higher than the radial thermal conductivity and the thermal conductivity of the matrix, which makes the heat transfer in the plate along the fiber axial direction significantly higher than the other directions. The fiber, like a heat transfer channel, allows more heat to be transferred from the high-temperature region to the low-temperature region, which intensifies the difference in heat transfer characteristics between fibers and matrix. Under the coupling effect of the above two factors, the overall cooling effectiveness of the plate fluctuates.

The fluctuation has different characteristics under different braided structures around the film-cooling hole. As shown in Figure 11b, the $\eta$ of case 1 and case 2 on line 1 gradually decreases along the flow direction, while the $\eta$ of case 3 and case 4 on line 1 decreases with fluctuation along the flow direction, and the fluctuation amplitude decreases gradually along the flow direction. The $\eta$ of case 1 and case 2 on line 2 is asymmetric, which is aggravated by the drill structure (case2). The difference between the two peaks of $\eta$ on

line 2 is up to 5.88%. The $\eta$ of case 1 and case 2 on line 2 is symmetrical concerning the centerline (X = 0). In general, the model with a drilled hole has slightly higher cooling effectiveness comparing with the model with a preformed hole. The average $\eta$ of Cases 1–4 on line 2 are 0.363, 0.366, 0.356 and 0.367, respectively. The difference between case 1 and case 2 is not obvious, while the $\eta$ of case 4 is higher by 3.1% compared with case 3.

The line 1 of case 1 and case 2 are located between the two weft yarns, so the temperature on the centerline is not affected by the fluctuation information. As shown in Figure 11c, the braided weft yarns of case 1 and case 2 are not symmetrically distributed along the central section of the film-cooling hole, which makes the $\eta$ of case 1 and case 2 asymmetric on line 2. Meanwhile, compared with the preformed structure, the drilled structure makes the fiber cross-section directly connected with the cooling flow. The flow will take more heat away, which makes the temperature fluctuation on line 2 more intense, and the overall cooling effect is slightly higher than that of the preformed structure on the hot-side wall.

### 3.1.2. Temperature Gradient Distribution of CCR

In practical engineering applications, the temperature distribution of the component has an important influence on structural stability. The temperature on the hot-side wall presents a fluctuation related to the braided structure, which is consistent with the results of the fluctuation distribution of the overall cooling efficiency on the hot-side wall described in Section 3.1.1. It can be found from Figure 11a that temperature fluctuation characteristics are different in different regions. In the upstream region ($-7.5D < Y < -0.5D$) and non-film-coverage of the downstream region ($-0.5D < Y < 15.5D$), the temperature of the matrix near the braided fibers is lower than the region without fibers at the same Y coordinate position. In the film coverage region, the opposite is true. Besides, from Figure 11a, it can be found that the different hole-forming methods mainly cause the temperature of the area around the film-cooling hole to change. To study the influence of different braided structures around the film-cooling hole on the temperature field and heat transfer path of the plate, a cuboid characteristic region (CCR) centered on the center of the film-cooling hole is selected. Figure 12a shows the temperature distribution of CCR. It can be found that the fiber temperature around EP-Hole and WP-Hole is higher than the fiber temperature around the corresponding DR-Hole. The drilling structure makes the cut-off fiber surface directly exposed to the cooling flow, and the fiber section directly contacts with the cooling flow to take away more heat, which makes the temperature of the fiber around the drilling structure lower than that of the braided fiber around the reserved hole.

As shown in Figure 12b, in the upstream region of the film-cooling hole, heat flux is injected from the upstream high-temperature region into the downstream low-temperature region. Due to the high heat transmission of the fiber, the fiber takes more heat to the downstream region, which makes the temperature of the area near the braided fiber upstream lower than in other regions. In the film coverage region, the heat is mainly transferred from the surrounding high-temperature solid region through thermal conduction. The region where the braided fiber is has more heat injected, which makes the temperature of the region where the fiber is higher than the surrounding region.

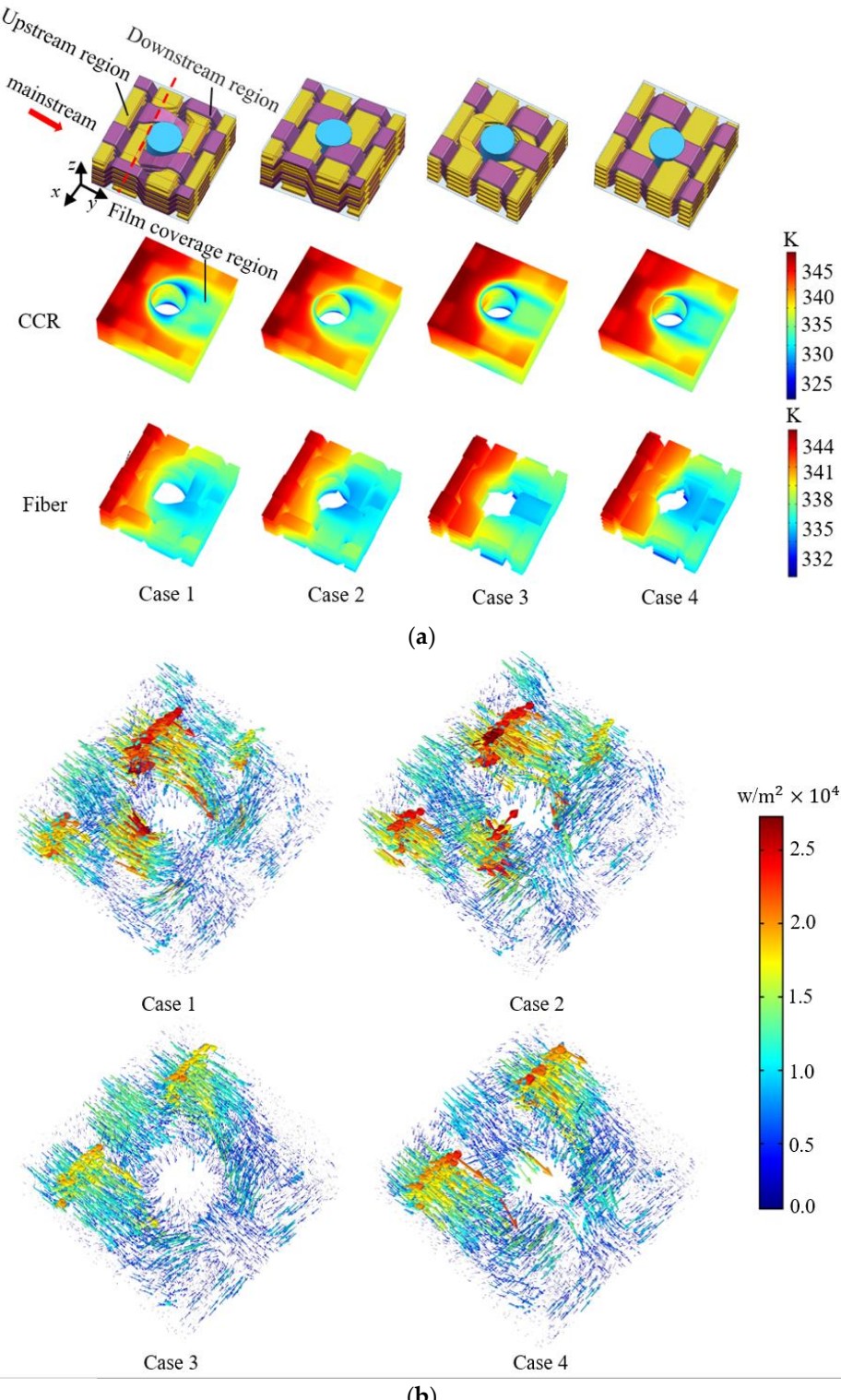

**Figure 12.** Temperature fields and heat flux direction of different cases on CCR: (**a**) temperature fields of different cases on CCR, (**b**) flux direction of different cases on CCR.

To further analyze the differences between the four models, Figure 13a shows the temperature distribution of cases 1−4 on sections 1 to 3. It can be found that the temperature distribution shows an obvious correlation with the braided structure. For the EP-Hole structure and DR-Hole 1 structure, the temperatures of section 1 and section 3 present asymmetric distribution. For WP-Holes and DR-Hole 2 structures, the temperature of sections 1 and 3 is distributed symmetrically. Comparing the four kinds of film-cooling

hole structures, it can be seen that the surrounding temperature of WE-Hole is lower than the other three structures. The influence of different hole-forming methods on the internal temperature of the plate is mainly concentrated in the range of $-1D < Y < 1.5D$ around the film-cooling hole. In the position of $Y < -1D$ upstream of the film hole, the average dimensionless temperature $\delta$ change caused by different hole-forming methods is less than 1%. In the position of $Y > 1.5D$ downstream of the film hole, the $\delta$ change caused by different hole-forming methods is less than 3%, as shown in Figure 13b. In the upstream of the film-cooling hole ($-1D < Y < -0.5D$), the $\delta$ of case 3 (WP-Hole) is about 0.74, which is not significantly different from case 4. The $\delta$ of case 1 (EP-Hole) and case 2 is about 0.68, and case 3 is 6% higher than case 2. Around the EP-Hole ($-0.25D < Y < 0.5D$), there is a low-temperature zone with alternating temperature along the thickness direction, and the minimum $\delta$ of case 1 in this region is 0.463, while around the WP-Hole ($-0.25D < Y < 0.5D$), there is a connected low-temperature zone along the thickness direction, and the minimum $\delta$ of case 3 in this region is 0.376. Compared with the two corresponding preformed hole structures, the low-temperature zone of the drilled hole structures becomes narrower; the minimum $\delta$ of case 2 and case 4 in this region are 0.496 and 0.417, respectively. The maximum $\delta$ of the four models around the film-cooling hole is significantly different. The dimensionless temperature of case 2 is 12% higher than that of case 3. Case 3 (WP-Hole) is beneficial to obtain a better cooling effect around the film-cooling hole.

It can be seen from Figure 13a that for the EP-Hole structure and DR-Hole 1 structure, the weaving direction of the weft yarns around the film-cooling holes is exactly the opposite, which leads to the asymmetrical temperature distribution. The two preformed hole structures will cause the warp yarns around the film-cooling hole to concentrate on both sides, which causes the low-temperature area around the film-cooling hole of the two reserved hole structures to be wider than the drilled structure.

### 3.1.3. Thermal Gradient Distribution of CCR

The thermal gradient is an important factor that affects the internal thermal stress of the material, especially for the region with a large thermal gradient near the film-cooling hole. Figure 14a shows the thermal gradient distribution of CCR. It can be seen from Figure 14a that the high thermal gradient mainly exists in the region close to the film-cooling hole. The thermal gradient is low in the middle of the plate and high on both sides. By comparing the four models, it can be seen that the temperature gradient distribution trends of the four models are essentially the same, but compared with case 1 (EP-Hole) and case 3 (WP-Hole), the temperature gradients of case 2 and case 4 around the cut fiber increase sharply. Figure 14b,c show the average thermal gradient and maximum thermal gradient of cases 1–4; it can be seen that different preformed holes have little effect on the average thermal gradient around the film-cooling holes, but have a significant effect on the maximum thermal gradient. The drilled structure will significantly increase the thermal gradient inside the plate. The maximum $\varepsilon$ of case 1, case 2, case 3, and case 4 are 2.724, 4.23, 2.35, and 5.26, respectively. Case 2 is 55.3% higher than case 1, and the case 4 is 123.3% higher than case 3. The high temperature gradient is mainly distributed in the area between the broken fiber and the wall, as shown in Figure 14a. In summary, the EP-Hole structure could reduce the average thermal gradient around the flat film-cooling holes, and the WP-Hole could reduce the maximum thermal gradient.

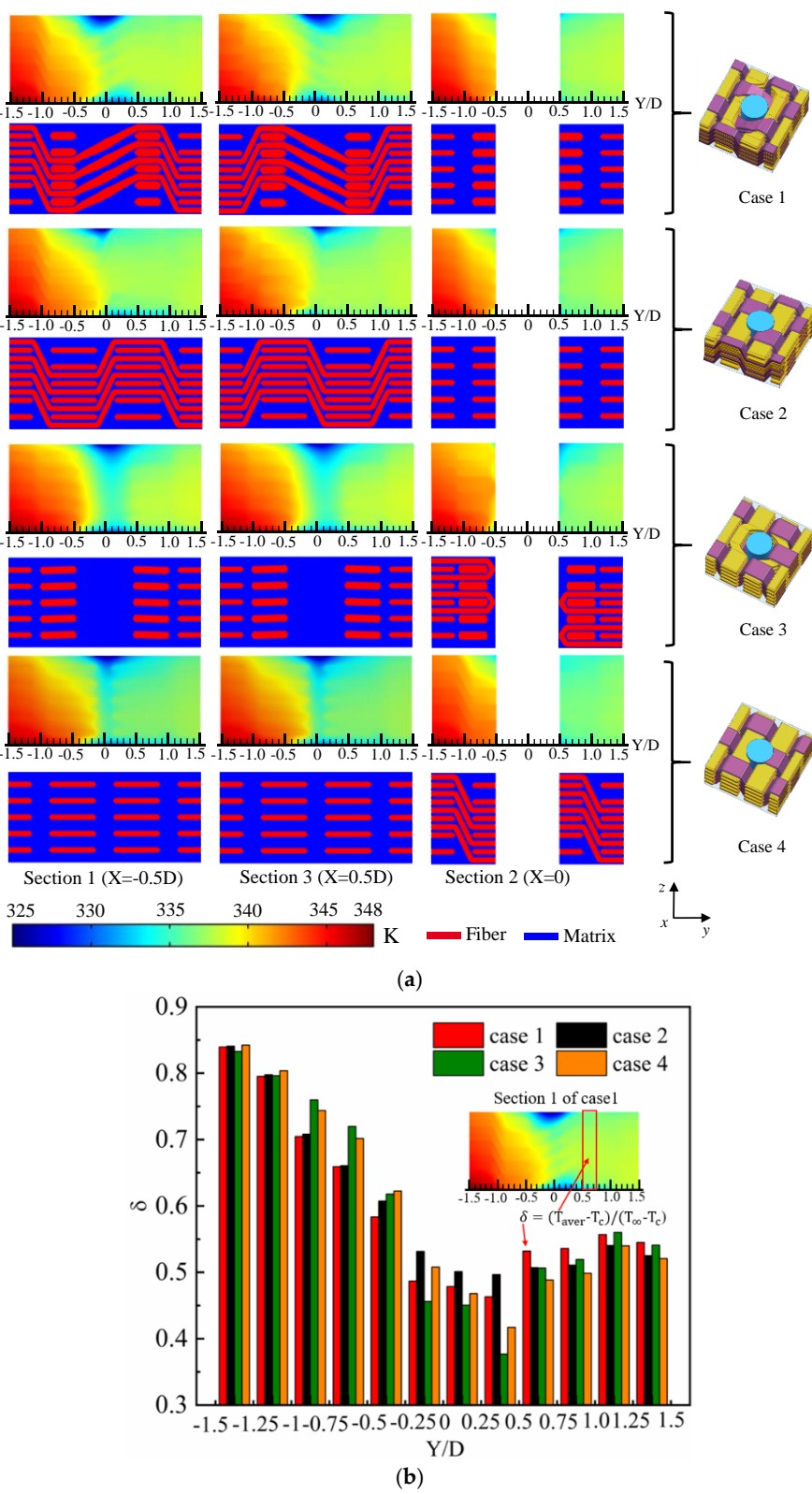

**Figure 13.** Temperature fields and heat flux direction of different cases on CCR: (**a**) temperature fields of different cases on sections 1 to 3, (**b**) average dimensionless temperature δ distribution of section 1.

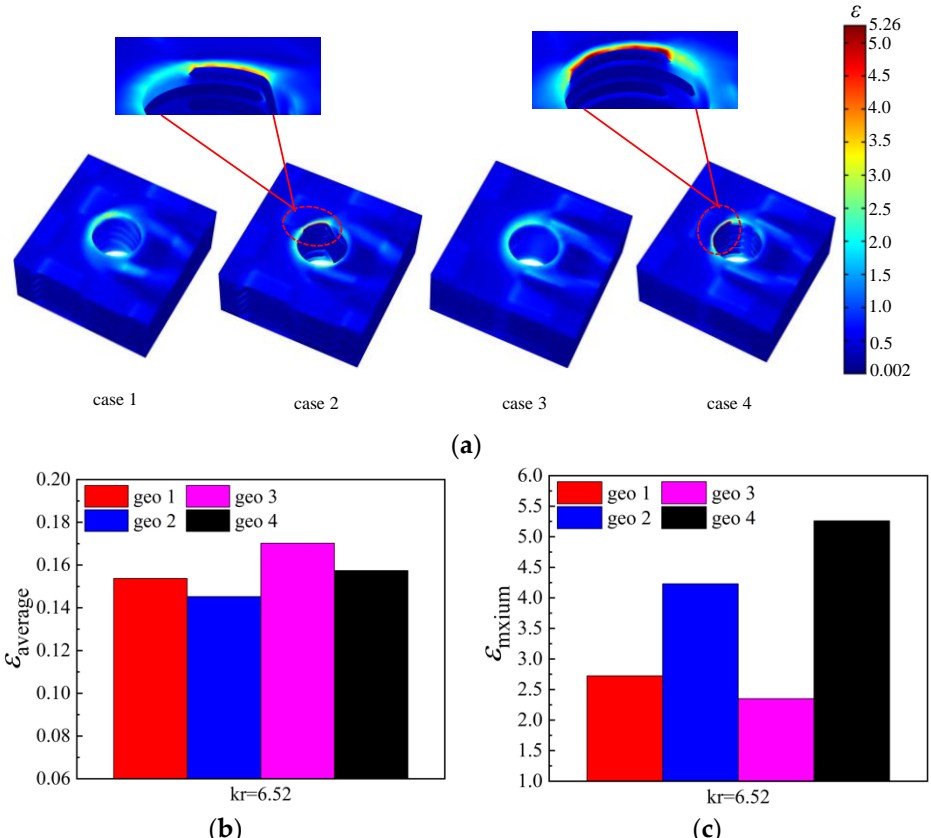

**Figure 14.** The thermal gradient of cases 1–4 on CCR: (**a**) thermal gradient of cases 1–4, (**b**) average relative temperature gradient, and (**c**) maximum relative temperature gradient.

The strong heat exchange of the CCR caused by the cooling flow is the main reason for the thermal gradient of the CCR. The film coverage region of the hot-side wall, the cold-side wall, and the film-cooling hole wall are directly in contact with the cooling flow, so the temperature near the wall is lower than that of the internal region, which makes the region close to these walls have a larger thermal gradient. The fiber directly contacts with the coolant flow in case 2 and case 4. Moreover, the thermal conductivity of the fiber is higher than that of the matrix, which makes the drilled structure (case 2 and case 4) in the same area able to take away more heat than the structure of the preformed hole (case 1 and case 3), resulting in a lower temperature and larger thermal gradient.

### 3.2. Influence of Kr

Figure 15a shows the cooling efficiencies of line 1 with different Kr. It can be found that with the increase of the axial thermal conductivity of the fiber, the $\eta$ of the plate decreases significantly in the region $0.5 \leq Y/D \leq 6$. When Kr increased from 3.26 to 13.05, the average $\eta$ of 0.5~6D on line 1 of the plate with the EP-Hole structure decreased from 0.37 (case 5) to 0.336 (case 9), a decrease of 10.12%. For the plate with the WP-Holes, the average η of 0.5–6D on line 1 is reduced from 0.358 (case 7) to 0.325 (case 11), which is also a decrease of 10.15%.

Figure 15b shows the heat flux on the hot-side wall and the film-cooling hole wall of the plates with EP-Hole and WP-Hole under different Kr values. It can be seen from Table 2 that as Kr increases from 3.26 to 13.05, the amount of heat entering the flat plate through the hot sidewall increases. For Geo1, the heat flux of the hot-side wall increased from 528.63 W/m² to 572.45 W/m², an increase of 8.29%. For Geo3, the heat flux of the hot-side wall increased from 526.25 W/m² to 564.96 W/m², an increase of 7.35%. With the increase of Kr, however, the heat taken away by convection heat transfer through the wall of the film-cooling hole also increased. For Geo1, the heat flux on the wall of the film-cooling

hole increased from 1813.1 W/m$^2$ to 2048.4 W/m$^2$, an increase of 12.97%. For Geo3, the heat flux on the wall of the film-cooling hole increased from 1744.4 W/m$^2$ to 1942.5 W/m$^2$, an increase of 11.35%. Therefore, with the increase of Kr, the decrease of η on the hot-side wall is not simply caused by the increase of convective heat transfer on the hot-side wall. Figure 15 shows the heat flux and temperature distribution in section 4. It can be seen from Figure 16a,b that as Kr increases, the heat transferred from the high-temperature region (upstream region) to the low-temperature region (downstream region) increases. The EP-Hole structure increased by 55.5%, and the WP-Hole structure increased by 88.23%. As the upstream injection heat increases, the temperature on section 4 also increases, as shown in Figure 16c. It can be seen that the heat injection increase from the upstream region to the downstream region is another important reason for the decrease of η. This also makes it so that in the downstream of the film-cooling hole, with the increase of Kr, the closer to the film-cooling hole, the more η decreases.

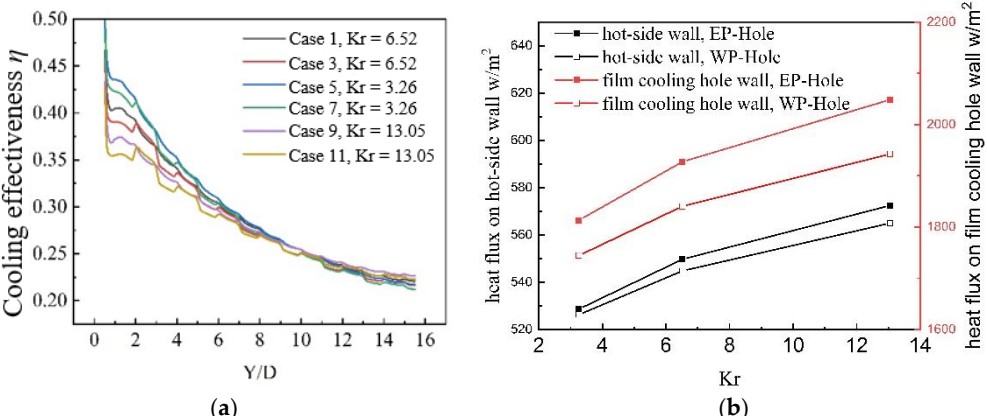

**Figure 15.** Cooling efficiencies and heat flux: (**a**) cooling efficiencies of line 1 with different Kr, and (**b**) heat flux on hot-side wall and film-cooling hole wall of EP-Hole and WP-Hole with different Kr.

Figure 16c shows that when Kr is low (Kr < 6.52), the WP-Hole structure can form a connected low-temperature zone between the cold sidewall, the hot-side wall, and the film-cooling-hole wall. However, the EP-Hole structure is unable to form such a connected zone. As Kr increases, the high-temperature zone of the WP-Hole structure is too large, resulting in the low-temperature-connected zone being cut off. As for the EP-Hole structure, the temperature on the film-cooling-hole wall always presents a distribution of high middle and low sides without the connected zone, due to the weft yarns.

As shown in Figure 17a, in terms of the average thermal gradient, different braided structures have little effect on the average temperature of the plate, compared with the effect caused by Kr. The changes brought about by different braided structures are less than 3.5%, but the average thermal gradient gradually decreases with the increase of Kr, as Geo1–4 decreased by 28.1%, 33.4%, 18.4%, and 25.4%, respectively. Regarding the maximum thermal gradient of the plate, the maximum thermal gradient change is less than 3% for the EP-Hole structure (Geo1) and WP-hole structure (Geo3) caused by different Kr. However, Kr has a significant effect on the maximum thermal gradient of the drilled structure. The increase of Kr will increase the maximum thermal gradient. The maximum thermal gradients for Geo2 and Geo4 increase by 10.7% and 18.8%, respectively, as shown in Figure 17b.

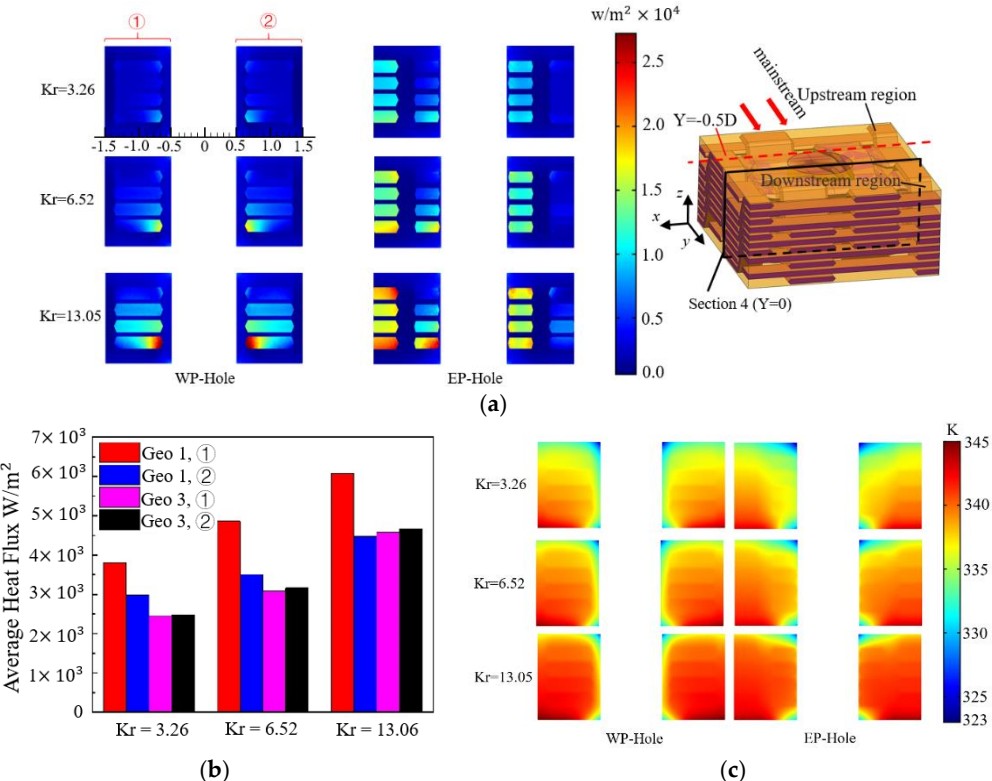

**Figure 16.** Heat flux and temperature field on section 4 of the different cases: (**a**) heat flux at Y component on section 4 of different cases, (**b**) histogram of average heat flux, and (**c**) temperature field on section 4 of different cases.

Different film-cooling holes only affect the local braided structure around them, so they have little effect on the average thermal gradient of the plate. With the increase of Kr, the high-temperature region expands, as Figure 16c shows, and the temperature around the film cooling hole becomes more uniform, so the value of the high thermal gradient region decreases. As stated in the above article, the drilled structure will cause more heat to be taken away from the wall of the film-cooling pores, which will result in a larger thermal gradient around DP-Hole than EP-Hole and WP-Hole. At the same time, with the increase of Kr, the heat exchange brought by the drilling becomes stronger, which in turn causes a higher thermal gradient. The preformed hole structure will not cause violent heat exchange around the film-cooling hole because the fiber does not directly contact the cooling flow, so Kr will not affect the maximum thermal gradient of the plate.

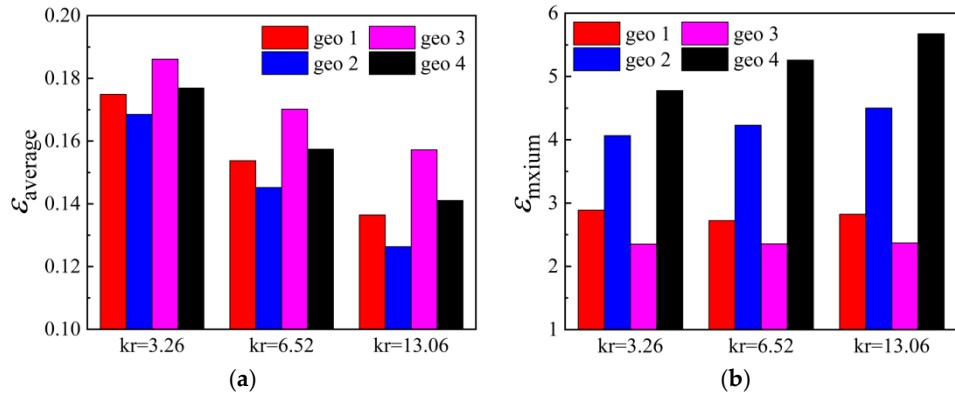

**Figure 17.** Histogram of the thermal gradient with different cases: (**a**) $\varepsilon_{average}$ of CCR with different Kr, and (**b**) $\varepsilon_{maximum}$ of CCR with different Kr.

## 4. Conclusions

In this paper, the influence of different preformed holes on the film-cooling performance over 2.5D braided CMC plates is numerically studied. Four models including EP-Hole, WP-Hole, and two types of DR-Hole are established. The distributions of temperature, thermal gradient, and cooling effectiveness based on these four models are compared and analyzed in detail. In addition, the ratio between the axial and radial thermal conductivity of fiber $K_r$ is changed, and its influence on the effectiveness of the cooling performance and internal temperature fields is studied. The conclusions could be drawn as follows:

1.  Different preformed holes have a significant impact on the temperature distribution inside the plate around the film-cooling hole; the change of dimensionless temperature inside the plate for the four models is up to 12%. The temperature of CCR in the models with EP-Hole and WP-Hole is lower than that of the model with the drilled hole. Furthermore, the WP-Hole forms a connected low-temperature area around the film-cooling hole. Compared with the drilled hole, the preformed holes can effectively reduce the maximum thermal gradient inside the plate. The maximum thermal gradient around the DP-Hole is significantly higher than that of WP-Hole and EP-Hole (DP-Hole 1 is 55.3% higher than EP-Hole and DP-Hole 2 is 123.3% higher than WP-Hole).

2.  With $K_r$ increasing, the overall cooling effectiveness of the 2.5D braided plate decreases. With the increase of $K_r$, the variation trend of the overall cooling efficiency of the plate brought by the four kinds of film holes is consistent. When $K_r$ increases from 3.26 to 13.05, the average overall cooling effectiveness dropped by about 10%. The difference in overall cooling efficiency caused by different preformed holes will not change much with the change of $K_r$.

3.  With the increase of $K_r$, the advantages of the two preformed hole structures in controlling the temperature gradient of the CMC plate with film-cooling holes are further expanded. The increase of $K_r$ has a little effect on the maximum thermal gradient of the preformed hole structure. The maximum thermal gradient change is less than 3% for the EP-Hole structure and WP-hole structure caused by the increase of $K_r$. However, the increase of $K_r$ has a significant effect on the maximum thermal gradient of the drilled structure. The increase of $K_r$ will increase the maximum thermal gradient. The maximum thermal gradients for DP-Hole 1 and DP-Hole 2 increase by 10.7% and 18.8%, respectively.

**Author Contributions:** Conceptualization, C.Z. and Z.T.; methodology, C.Z. and Z.T.; software, C.Z.; validation, C.Z., Z.T., and J.M.; formal analysis, C.Z.; investigation, C.Z. and Z.T.; resources, Z.T. and J.M.; data curation, Z.T. and J.M.; writing—original draft preparation, C.Z.; writing—review and editing, C.Z. and J.M.; visualization, J.M.; supervision, J.M.; project administration, J.M.; funding acquisition, J.M. All authors have read and agreed to the published version of the manuscript.

**Funding:** This research was funded by the National Natural Science Foundation of China, grant number 51906105; the Jiangsu Provincial Natural Science Foundation of China, grant number BK20190420; the China Postdoctoral Science Foundation, grant number 2018M642248; and the National Science and Technology Major Project, grant number 2017-III-0003-0027.

**Institutional Review Board Statement:** Not applicable.

**Informed Consent Statement:** Not applicable.

**Data Availability Statement:** Data are contained within the article.

**Acknowledgments:** The authors gratefully acknowledge the financial support for this project from the National Natural Science Foundation of China, grant number 51906105; the Jiangsu Provincial Natural Science Foundation of China, grant number BK20190420; the China Postdoctoral Science Foundation, grant number 2018M642248; and the National Science and Technology Major Project, grant number 2017-III-0003-0027.

**Conflicts of Interest:** All the authors certify that this paper described original research that has not been published previously, and is not under consideration for publication elsewhere, in whole or in part. No conflict of interest exists in the submission of this manuscript. All the authors listed have approved the enclosed manuscript.

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
