# Peer review of "Investigation of the Film-Cooling Performance of 2.5D Braided Ceramic Matrix Composite Plates with Preformed Hole"

_aerospace, doi:10.3390/aerospace8040116_

Round 1

Reviewer 1 Report

The manuscript is interesting and could be valuable for the special issue but the Authors have not avoided a few important shortcomings:

1) Blowing ratio described by Eq3 is 0.25 but the Authors have chosen [34] for reference in which the ratio is 1.5. Why did you choose this paper? What is the point?

2) Furthermore, [34] is the numerical benchmark and not the experimental result.

3) I do not see the meaning of Eq7. The gradient is valuable by itself. Why defining another parameter?

4) Section 2.2: assumption of two-dimensional anisotropy of thermal conductivity is sufficient and Eq4 is enough.

5) Assumption of turbulence intensity of 0.2% should be explained deeper.

6) Table 2. It is rather hard to read because the data in the table are scattered. 

Author Response

We are very grateful for the reviewer’s enlightening and insightful comments. Please see the attachment for detailed reply

Reviewer 2 Report

Dear authors,

This is indeed a niece piece of research. I suggest to carry on in this research line.

The Reviewer

Author Response

We are very grateful for reviewer's affirmation.We will carry on in this research line.

Reviewer 3 Report

Paper entitled “Investigation of the Film Cooling Performance of 2.5D Braided Ceramic Matrix Composite Plates with Preformed Hole” meets the necessary standards for publication in this journal.

I recommend: Be careful when writing the section numbers and titles.

Author Response

We are very grateful to the reviewers for your careful reading and examination of this article. We have carefully examined the section numbers and titles. We found an error in line 230 of article, and we have modified, which should be Section 2.3 of the article.

Round 2

Reviewer 1 Report

Thank you for the Authors' response. The Authors significantly corrected the manuscript, and remarks were exlained.